

# Bioremediation of engine-oil contaminated soil using local residual organic matter

Kawina Robichaud[1], Miriam Lebeau[2], Sylvain Martineau[2] and Marc Amyot[1]

[1] Département de sciences biologiques, Université de Montréal, Montréal, QC, Canada
[2] Akifer, Boucherville, QC, Canada

## ABSTRACT

Soil remediation industries continue to seek technologies to speed-up treatment and reduce operating costs. Some processes are energy intensive and, in some cases, transport can be the main source of carbon emissions. Residual fertilizing materials (RFM), such as organic residues, have the potential to be beneficial bioremediation agents. Following a circular economy framework, we investigated the feasibility of sourcing RFMs locally to reduce transport and assess possible bioremediation efficiency gains. RFMs were recruited within 100 km of the treatment site: ramial chipped wood (RCW), horse manure (MANR) and brewer spent grain (BSG). They were added to the land treatment unit's baseline fertilizer treatment (FERT, "F") to measure if they improved the remediation efficiency of an engine oil-contaminated soil ($7,500 \pm 100$ mg kg$^{-1}$). Results indicate that MANR-F was the only amendment more effective than FERT for petroleum hydrocarbons (PHC) reduction, while emitting the least $CO_2$ overall. RCW-F was equivalent to FERT but retained more moisture. Although BSG contributed the most nitrogen to the soil, BSG-F retained excessive moisture, emitted more volatile organic compounds, contained less soil $O_2$, and was less effective than the baseline treatment. Significantly more of the $C_{16}$–$C_{22}$ fraction was removed ($63\% \pm 22\%$) than all other fractions ($C_{22}$–$C_{28}$, $C_{28}$–$C_{34}$, $C_{34}$–$C_{40}$), which were equally removed. Microbial community-level physiological profiling was conducted with Biolog Ecoplates$^{TM}$, and catabolic diversity differed between treatments (utilization rates of 31 carbon sources). MANR-F has the potential to increase PHC-remediation speed and efficiency compared to inorganic fertilizer alone. Other RFM promote moisture retention and diverse microbial catabolic activity. A variety of RFM are present across the globe and some can offer low-cost amendments to boost remediation efficiency, while reducing treatment time compared to traditional fertilizer-only methods.

## INTRODUCTION

Soil treatment centers are constantly seeking new soil remediation technologies to speed up treatment and reduce their production costs. Conversely, organic waste diversion from landfills and processing strategies for sustainable waste management practices are actively being put in place across Europe and North America (*Commission for*

Corresponding author
Marc Amyot,
m.amyot@umontreal.ca

*Environmental Cooperation (CEC), 2017*; *European Commission, 2010*). Organic waste makes up a significant portion of the waste stream going to landfills where its anaerobic decomposition produces polluting gases, like methane, which not only affect human health and air quality, but also contribute to climate change (*Lou & Nair, 2009*). In Canada, all levels of government aim to reduce greenhouse gas emissions, namely through the reduction of putrescible landfill waste (*Environment Canada, 2014*). For example, the Quebec Ministry of Environment and Fight Against Climate Change (MELCC) encourages the valorization of residual fertilizing materials (RFM) to divert them from landfills where they become putrescible organic residues. These include, but are not limited to, residential food scraps, agricultural and industrial wastes and wood residues. This diversion, which reached 1.6 million tons in 2012, can be used to revegetate degraded sites and serve as fertilizer in agriculture, thereby helping to reduce greenhouse gas emissions (*Larose, Hébert & Plante, 2014*). Several studies demonstrate the potential of RFM as efficient bioremediation agents when compared to controls (*Abioye, 2011*; *Cole, 1998*; *Guidi, Kadri & Labrecque, 2012*; *Hupe et al., 1996*; *Margesin & Schinner, 1997*; *Onwosi et al., 2017*; *Zubillaga, Bressan & Lavado, 2012*). However, the *Larose, Hébert & Plante (2014)* reports that only 8% of salvaged RFM were used on degraded sites and usage on soils contaminated with petroleum hydrocarbons (PHC) was not mentioned. We reported the current practices in the province of Quebec, where our project takes place, but the main principles of organic waste reduction apply across Canada, the US, Mexico and Europe (*Commission for Environmental Cooperation (CEC), 2017*; *European Commission, 2010*).

Further, these main principles are embedded into a circular economy framework. Circular economy is an emerging concept which considers how we can consume goods and services without depending on the extraction of raw natural resources, thereby closing loops that prevent the disposal of materials in landfills (*Sauvé, Normandin & Mcdonald, 2016*). In contrast to the usual "linear economy" model, the impacts of resource consumption are taken into account. Governments around the world are adopting the concept of circular economy (e.g., China and Europe; see *Sauvé, Bernard & Sloan, 2015*). This concept applies to soil remediation practices because some remediation approaches are energy-intensive or require large areas of land (landfarming, soil vapor extraction, thermal desorption, etc.) and many depend on the introduction of inorganic fertilizers which rely on energy-intensive synthesis and mining of non-renewable resources, such as phosphorus (*Daneshgar et al., 2018*; *Khan, Husain & Hejazi, 2004*; *Rafiqul et al., 2005*).

A popular soil remediation approach to large-scale PHC-contamination is the use of low-tech and relatively low-cost biopiles (*Ivshina et al., 2015*). This remediation method is normally based on the stimulation of endemic microbes for contaminant degradation (*Khan, Husain & Hejazi, 2004*). It is optimized through different manipulations such as the addition of nutrients, the modification of soil structure and exerting some control over moisture content and air supply (*Juwarkar, Singh & Mudhoo, 2010*). In Quebec, regulations require that biopiles must be covered and equipped with a pulled-air system and an air treatment system for volatile organic carbon contaminants, like BTEX (benzene, toluene, ethylbenzene and xylenes), methane, hexane and other light PHC molecules having less than 12 carbon atoms ($C_{12}$), prior to release in the atmosphere.

Appropriate nutrient input and timing of application are key for effective bioremediation of PHC-contaminated soils (*Nwankwegu, Orji & Onwosi, 2016*). The main sources for nutrients are inorganic fertilizer and organic matter amendments to the contaminated soil. Nitrogen is the most common limiting nutrient, but depending on its form, it may or may not be available for the microorganisms in the soil that biodegrade contaminants (*Schulten & Schnitzer, 1997*). However, accurately monitoring different forms of N is sometimes difficult because the microbe-moderated nitrogen cycle in the soil is rapid and complex. Microbes are at the heart of bioremediation techniques like biopiles and it is advantageous to understand how soil amendments may influence them. Community-level physiological profiling (CLPP) is an approach commonly used in ecology for assessing microbial community profiles in soils (*Jones et al., 2018*). Biolog Ecoplates[TM] give a sensitive fingerprinting tool for communities' catabolic diversity. The microplates contain 31 different carbon sources, along with a redox dye which turns purple when the substrate is consumed (*Biolog, 2018*). Based on a specific soil's heterotrophic bacterial community, different carbon sources are consumed at varying rates, offering a unique look at the functional carbon use in a given soil, which can lead to the statistical differentiation of soils based on Ecoplates[TM] data alone (*Jones et al., 2018*). The method's limits include the issue of poor laboratory cultivability of certain bacterial strains and the challenges associated with the huge amount of data generated. Nonetheless, the data obtained is information-rich and reproducible (*Bradley, Shipley & Beaulieu, 2006*).

For this study, we aimed to find three close-proximity RFM which could be further valorized as biological agents to stimulate the biodegradation of PHCs. (1) Wood residues have been used as bulking agents in bioremediation projects (*Battelle & NFESC, 1996*; *Kauppi, Sinkkonen & Romantschuk, 2011*). We opted for ramial chipped wood (RCW), a material that has been proven to increase soil fertility (*Lemieux, 1986*). Its chemical composition has rich ratio of polysaccharides to proteins (C:N) varying between 50:1 and 175:1 (varying across species and seasons) compared to woodchips from stem wood which have a C:N ratio of 400:1 to 600:1 under the same conditions (*Lemieux, 1986*). RCW is rarely cited in bioremediation, but *Hattab et al. (2015)* found that it reduced the toxicity (determined by mobility and phytoavailability) of trace metals in plants. (2) Animal manure can contain high levels of nutrients such as N, P and K (*Moreno-Caselles et al., 2002*). *Kirchmann & Ewnetu (1998)* found that co-composting with horse manure (MANR) could reduce the concentration of large paraffin molecules by 80% in 110 days, as well as 93% of petroleum residues in 50 days. *Cole (1998)* reported that mature manure (6 months) was efficient and reduced treatment time from the typical 6 months or more to 2 months or less for oily sludges (PHC in the engine oils molecular range). (3) Brewers' spent grain (BSG), which is the mass of wet grains remaining after the beer manufacturing process, is a fast-growing RFM in our region. The number of micro-breweries has more than doubled (105 new) in the last 7 years (*Association des Médecins Biochimistes du Québec (AMBQ), 2018*). The production of BSG for micro-breweries in Quebec is approximately 970,000 kg year$^{-1}$. BSG is rich in nutrients and contains 77–81% water (w w$^{-1}$) (*Santos et al., 2003*). *Abioye (2011)* suggests that BSG is a promising agent for the bioremediation of soil contaminated with PHC and metals.

Research by *Santos et al. (2003)* shows that the BSG resulting from different types of beer within the same brewery was fairly homogenous, thus ensuring greater replicability.

In bioremediation, laboratory results do not always correspond with field observations since the dynamics of large-scale bioremediation systems are different. Key soil parameters (such as moisture, aeration, temperature, homogeneity, etc.) become more difficult to control with increasing size, and bioremediation dynamics do not seem to follow a linear transfer in efficiency from small to medium to large scale (*Khan, Zytner & Feng, 2015*; *Ko et al., 2007*). Furthermore, the legal regulations for industrial applications do not need to be respected in laboratory experiments. The scaling up of experiments must respect the legal framework applicable to the region or go through a process of application for approval, which can slow down technology transfer. In this study, we worked in partnership with an industrial soil remediation operator (Akifer for the SolNeuf treatment site) who projected to treat heavy PHC fractions (PHC molecules with more than 20 carbon atoms ($C_{20}$)) on their platform. The remediation target for the contaminant chosen (engine oil) was set to the province of Quebec's "C" criteria (commercial use, less than 3,500 mg kg$^{-1}$ for the PHC molecules in the $C_{10}$–$C_{50}$ range), and was to be achieved within one treatment season in pilot scale (0.76 m$^3$) experimental units, using different local residual organic matter found close to the treatment platform. We aimed to quantify if there was preferential degradation of some PHC fractions among treatments. We sought to determine how the amendments changed the metabolic activity of the soil's bacterial communities and if distinctive "fingerprints" could be identified.

## MATERIALS AND METHODS

### Soil preparation

This experiment was conducted on a certified treatment platform for contaminated soils owned by SolNeuf Inc., and operated by AKIFER Inc., located in Neuville, QC, Canada (46°44′5.0352″N, 71°41′2.634″W). The soil used was sandy (98.4% ± 0.3% sand, 2.5% ± 0.2% gravel, 0.9% ± 0.1% silt and loam), with a pH of 6.4 (1:1, soil:$H_2O$). It contained less than 0.3% organic matter (combustion) and 34.4 mg kg$^{-1}$ phosphorus (P). The soil was free of PHC as measured by gas chromatography with flame ionization detector (GC-FID) according to the method MA. 400—HYD. 1.1 (*Centre d'expertise en analyse environnementale du Québec (CEAEQ), 2016*). In order to control the concentration and homogeneity of the contamination, the initial soil was artificially contaminated with unused engine oil to 7,500 mg kg$^{-1}$ of PHC ($C_{10}$–$C_{50}$) and thoroughly mixed for 1h by excavator. The oil was RUBIA LD 10W30, manufactured by Total for diesel motors. The PHC concentrations were followed from June to November 2015. The target for remediation was the Government of Quebec's "C" Criteria (less than 3,500 mg kg$^{-1}$ in the $C_{10}$–$C_{50}$ range) (*Beaulieu, 2016*).

### RFM soil amendments

For the RFM materials, we defined as "local" the ones that originated from a distance of 100 km or less of the soil treatment platform. This action was guided by a circular economy framework and the *Interstate Technology & Regulatory Council (ITRC) (2011)*

Regulatory Guidance for Green and Sustainable Remediation to minimize the project's carbon footprint by reducing fuel consumption associated with transport. A recycling and triage center located just 600 m from the treatment platform was the first location where RFM were sought out. RCW of mixed origin (unspecified mix of deciduous trees and conifers, excluding *Thuja* sp.) was chosen. RCW differs from regular wood chips because it originates from branches less than seven cm in diameter and not whole trunks (stem wood) (*Lemieux, 1986*). The second closest available RFM was excess MANR from stables eight km away (homogeneous mix of fresh to 8 months old manure). The last RFM was BSG. Initially intended to be sourced 12 km away, the BSG used for the project had to be sourced 92 km away because of time constraints. The spent grain used in this project was 4 days old and had been stored outdoors in barrels.

## Experimental design

The four treatments used were: 1—Ramial Chipped Wood (**RCW-F**), 2—Brewer's Spent Grain (**BSG-F**), 3—Horse manure (**MANR-F**), and 4—Fertilizer alone (**FERT**) (88.6% calcium mononitrate (27 N:2.7 Mg:4.6 Ca) and 11.4% diammonium phosphate (18 N:46 P:0 K) to achieve a C:N:P ratio of 100:2.5:0.5). This fertilizer dosage is the usual treatment at this treatment platform, and it was also added as a nutrient baseline to all other treatments ("F"). Due to space and logistical constraints, no bins were tested without fertilizer. The organic amendments were added at a ratio of 30% by volume for each local organic amendment and were mixed for 10 min by excavator, followed by 10 min of hand-shovel mixing (see Fig S1 for design). For each treatment, three concrete cylinders (0.76 m$^3$) were placed outdoor and filled with 760 L of the resulting mixtures of soil and amendments. The 12 cylinders were randomly distributed in a straight-line oriented southwest-northeast. Following the province of Quebec's regulations, the tops of the bins were covered with a thick plastic and a pulled-air system was connected to a perforated 2″ PVC pipe placed 48 cm from the bottom of each bin (10% below the center). The air was drawn at a rate of $1.5 \pm 0.26$ m$^3$ h$^{-1}$ and passed through a biofilter prior to release to the atmosphere. The rate was set to be proportional to the airflow in large biopiles on this site. Finally, all water runoff was collected and sent for treatment at a registered facility. The targeted soil moisture concentration was set at 70% of field capacity. When the moisture levels dropped significantly below that target, watering events took place ($n = 3$).

## Sampling

The concentrations of $C_{10}$–$C_{50}$ PHC in the soil were monitored three times over 5 months, on June 10, August 31 and November 2, 2015. On the first and last sampling dates, the $C_{10}$–$C_{50}$ PHC fractions were split into six smaller fractions ($C_{10}$–$C_{16}$, $C_{16}$–$C_{22}$, $C_{22}$–$C_{28}$, $C_{28}$–$C_{34}$, $C_{34}$–$C_{40}$, and $C_{40}$–$C_{50}$) to quantify if there was preferential fraction degradation. At each sampling event two new holes were drilled with a manual auger (five cm diameter). Samples were taken from 30 and 60 cm depths in the first hole, and from 60 to 90 cm depths in the second hole. These four samples were homogenized and sub-sampled for all analyses. Nutrient measurements were conducted from the first and last sampling events described above. For moisture readings, the soil was sampled

eight times and was measured gravimetrically (oven dry). Gas measurements were taken from a tube linked to a perforated PVC pipe capped on both ends and placed in the center of each bin. Carbon dioxide ($CO_2$), oxygen ($O_2$), and volatile organic compounds (VOCs) were measured with a portable gas detector (Eagle model; RKI Instruments, Union City, USA). The tube's end was equipped with a clamp, which was released to measure the gases present in the bins. Temperature loggers (Levelogger Gold model 3001; Solinst, Georgetown, Canada) were placed inside the PVC pipes used for gas monitoring in each experimental unit. Temperature was recorded every hour throughout the experiment. The loggers were calibrated at the beginning of the experiment and the deviation between loggers did not exceed 0.3 °C (0.14 ± 0.03 °C (air) and 0.08 ± 0.03 °C (water)).

## Analytical methods

The PHC fractions were measured by GCFID according to the MA. 400—HYD. 1.1 method (*Centre d'expertise en analyse environnementale du Québec (CEAEQ), 2016*). Soil samples were first dried with acetone (CAS no 67-64-1) and then extracted with hexane (CAS no 110-54-3) using a "paint mixer" type extraction system. Subsequently, silica gel (60–200 mesh grade 62 (CAS no 112926-00-8), $SiO_2$) was added to the extract to adsorb the polar substances. Finally, the supernatant hexane was analyzed by GC-FID. The concentration of hydrocarbons present in the sample was determined by comparing the total area of all peaks from n-$C_{10}$ to n-$C_{50}$ with the surfaces of the standards used to establish the calibration curve under the same assay conditions (diesel standard solution altered to 50% at 5,000 μg mL$^{-1}$ (Diesel fuel No. 2; Restek, Bellefonte, PA, USA). The methodological limit of quantification was 100 mg kg$^{-1}$ and PHC recovery rates were all between 82% and 94%.

The determination of ammoniacal nitrogen was a two-step process conducted according to method MA 300-N 2.0 (*Centre d'expertise en analyse environnementale du Québec (CEAEQ), 2014a*). The first step was an extraction in the presence of potassium chloride (CAS no 7747-40-7). Secondly, the ammonium ion reacted with sodium salicylate (CAS #54-21-7), nitroferricyanide (CAS no 13755-38-9) and dichloroisocyanuric acid (CAS no 2893-78-9) to form a blue alkaline complex with an absorbance at 660 nm, which is proportional to the concentration of ammoniacal nitrogen. Ammoniacal nitrogen recovery rates were between 95% and 96%. Phosphorus concentrations were determined according to method MA. 300—NTPT 2.0 (*Centre d'expertise en analyse environnementale du Québec (CEAEQ), 2014b*). Nitrate and nitrite ions were analyzed in accordance with method MA. 300—Ions 1.3 (*Centre d'expertise en analyse environnementale du Québec (CEAEQ), 2014c*). The soil was mixed with water in order to dissolve the extractable anions. For leached nitrates and nitrites, the extraction was done with the leaching buffer as specified in the Hazardous Materials Regulations and described in method MA. 100—Lix.com. 1.1 (*Centre d'expertise en analyse environnementale du Québec (CEAEQ), 2012*). Subsequently, anions are were separated by an ion exchange column using the following eluent solution: 0.0027M sodium carbonate (CAS no 497-19-8) and 0.0003M sodium bicarbonate (CAS no 144-55-8)). The retention time differs for each anion, which makes it possible to identify and dose them. Nitrates and nitrites were measured

using a conductivity sensor and the measured conductivity was proportional to the concentration of the anion in the sample. Recovery rates were between 97% and 104%.

Total phosphorus was determined in two steps. First, an acid digestion with sulfuric acid (CAS no 7664-93-9), which transforms phosphorus into orthophosphate. Secondly, orthophosphate ions were assayed by an automated system. The orthophosphate ion reacted with molybdate (CAS no 12054-85-2) and antimony (CAS no 28300-74-5) ions to form a phosphomolybdate complex. The latter was reduced with ascorbic acid (CAS no 50-81-7) to trigger the appearance of molybdenum blue, whose absorbance at 660 nm is proportional to the concentration of the orthophosphate ion. The detection limit was 200 mg kg$^{-1}$ and recovery rates were within ±15%.

Total organic carbon was measured by combustion according to method MA. 310-CS 1.0 (*Centre d'expertise en analyse environnementale du Québec (CEAEQ), 2013*) at 1,360 °C for a maximum of 600 s (oxygen at 30 lb in$^2$). Recovery rates were between 112% and 113%. The water used for the preparation of reagents and standard solutions was distilled or demineralized water. All samples were kept at 4 °C until analysis (less than 14 days).

### Community-level physiological profiling

The 96-well microplates contained 31 individual carbon sources from six classes (amine, amino acids, carbohydrates, carboxylic acids, phenolic compounds and polymers) (replicated three times in each plate) along with a clear tetrazolium dye, which gets irreversibly reduced to purple formazan dye when the bacteria consume the carbon (*Bochner, 1989*). One plate was used for each treatment cylinder ($n = 9$ carbon sources per treatment). A soil-aqueous solution was made according to methods previously described (*Choi & Dobbs, 1999*). All optical densities were brought below 0.350 nm (0.264 ± 0.009 nm) through dilutions to normalize the optical density at time zero. The well color development was monitored every day for 6 days by spectrophotometer (590 nm) and raw data was normalized by subtracting the control wells (water) present within each plate's three replicates. The result were interpreted by the rate of color change in the wells, the area under the curve and the functional richness as described in *Choi & Dobbs (1999)* and *Garland (1997)*. The average well color development (AWCD) was calculated with the following formula: $\text{AWCD} = \sum^{(C-R)} n$, where $C$ is color production, $R$ is the absorbance of the control (water) and $n$ is the number of substrates ($n = 31$). To independently estimate color development, a curve-integration (CI) approach was used. The area under the curve (the trapezoid area) was calculated as: $\sum_{i=1}^{n} \frac{v_i + v_{i-1}}{2} \times (t_i - t_{i-1})$ where $v$ is optical density at time $t$. Functional richness is defined by the number of positive wells ($\text{OD}_{\text{final}} - \text{OD}_{\text{initial}} > 0.25$ nm).

### Statistical analysis

Decreases in PHC concentrations, moisture levels, EcoPlates$^{\text{TM}}$ data, and temperature between treatments were assessed with a linear mixed-effects repeated measures model fitted by restricted maximum likelihood, performed with the nlme{} (*Pinheiro et al., 2017*) in *R Development Core Team (2016)*. Post hoc multiple comparison analysis were performed with the Tukey HSD method (lsmeans{} (*Lenth, 2016*) and mltcompview{}

(*Hothorn, Bretz & Westfall, 2008*)). Assumptions of normality and homoscedasticity were met for all mixed model tests. Gas readings ($O_2$, $CO_2$, VOC) were analyzed with the Kruskal–Wallis test by ranks for non-parametric data agricolae{} (*De Mendiburu, 2017*). Multiple comparisons were corrected with Bonferroni for all parametric and non-parametric tests. Figures were built with ggplot2 (*R Development Core Team, 2016*; *Wickham, 2009*).

## RESULTS

### Soil properties

In the first month, BSG-F temperatures in two bins spiked at least 14 °C above the other bins. Overall, the maximum temperatures reached were 45, 31, 29 and 28 °C for BSG-F, MANR-F, RCW-F and fertilizer treatment (FERT) respectively. There was variability between bins of the same treatment and between treatments, but for the first 90 days, the mean temperature across bins was 22.9 ± 2.7 °C ($n = 25,920$). Subsequently, the bins' internal temperatures gradually dropped with the advent of the fall season (Fig. 1A). At this point, BSG-F remained significantly higher than the other treatments (with the exception of the 10-day increment reported at 140 days) (mixed model, $p < 0.05$). The addition of engine oil (Total Base Number: 10) raised the initial soil pH by around one unit, to approximately 7.4 across all treatments. Subsequently, all treatments remained stable with a pH between 7.2 and 7.6, except for the BSG, (6.8 ± 0.4, after oil addition, and increased to 7.6 ± 0.2 by the end of the experiment). Water requirements varied between treatments. To maintain soil moisture near 70% of field capacity, three watering events took place during the course of the experiment. FERT had little water-holding capacity and needed watering each time. RCW-F required watering during the first two events, but not the third. MANR-F remained within a desirable moisture range (70% of field capacity) for the duration of the experiment and did not require any additional watering. Finally, BSG-F did not require any watering either and it contained the highest moisture levels of all treatments. There was a strong correlation between the amount of moisture (%) and the organic carbon content in the soil (excluding PHC-based carbon). (Fig. 1B).

### Nutrient balance

Raw BSG was the amendment richest in nitrogen whereas RCW contained the least nutrients (Table 1). All treatments experienced a sharp decline in available nitrogen ($NO_2^-$, $NO_3^-$ and $NH_4^+$), which was likely assimilated by soil microorganisms, with the exception of BSG-F, where a sharp increase in $NH_3$ and $NH_4^+$ was observed between June and November. Treatments all show relatively low total nitrogen loss (<15%), except BSG-F that lost nearly 25%. For FERT, an increase in total nitrogen was measured. Since there were no nitrogen inputs over the course of the project, this is probably linked to a heterogeneous distribution of the fertilizer in the soil.

### Engine oil decrease

A sharp decrease in the first 82 days, followed by a plateau was observed for all treatments, except the BSG, which maintained a slower but constant reduction over the 5-month
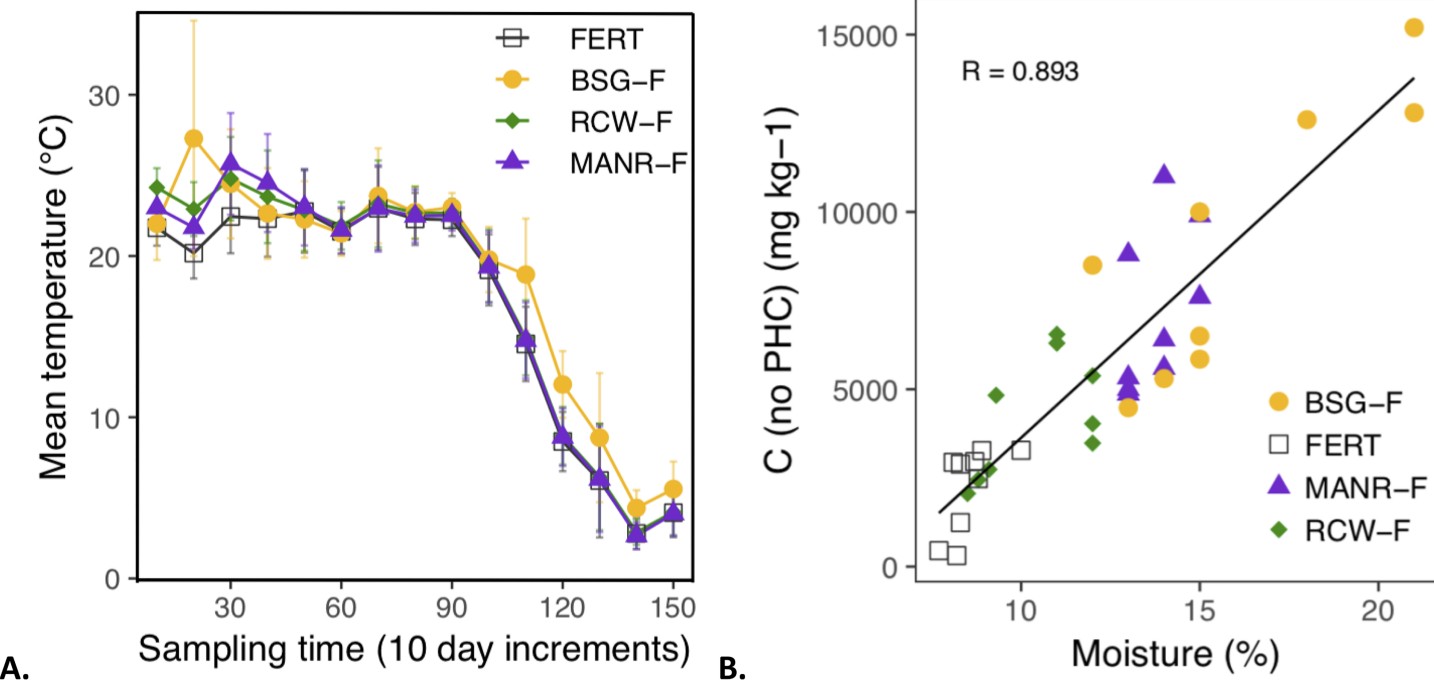

**Figure 1** **Soil temperature and moisture readings.** (A) Mean temperature in the center of the treatment bins in 10-day increments ($n = 720$). Error bars are SD. (B) Correlation ($R = 0.893$) between soil carbon and moisture (%) in the experimental bins from June, September and November data. The carbon contents attributable to PHC concentrations were removed, as they do not contribute to water retention in the soil.

**Table 1** Nutrient and organic matter content in raw amendments prior to incorporation in soil as well as measurements in the soil matrix at the beginning (June) and end of the experiment (November).

**Raw local organic ammendments**

| Source material | Analyte units | $NH_{3-4}$ mg kg$^{-1}$ | Total N mg kg$^{-1}$ | Total P mg kg$^{-1}$ | Total K mg kg$^{-1}$ | Org. matter % | C:N ratio |
|---|---|---|---|---|---|---|---|
| MANR | | 0.14 ± 0.02 | 2.8 ± 0.0 | 1.00 ± 0.01 | 4.75 ± 0.22 | 66 ± 2.5 | 36 ± 2.6 |
| RCW | | 0.20 ± 0.05 | 2.2 ± 0.75 | 0.21 ± 0.03 | 0.61 ± 0.10 | 85 ± 0.5 | 74 ± 20 |
| BSG | | 0.12 ± 0.02 | 5.8 ± 0.20 | 1.08 ± 0.03 | 0.39 ± 0.02 | 96 ± 0.0 | 18 ± 0.9 |

**Soil measurements**

| Treatment | Analyte units | $NH_{3-4}$ mg kg$^{-1}$ | $NO_{3-4}$ mg kg$^{-1}$ | Total N mg kg$^{-1}$ | TOC:N ratio | $C_{10}$–$C_{50}$ mg kg$^{-1}$ | Reduction %, rank |
|---|---|---|---|---|---|---|---|
| FERT | June | 61 ± 29 | 78 ± 8 | 196 ± 50 | 44:02.5 | 7 500 ± 100 | 62 ± 4[b] |
| | Nov. | 10 ± 00 | 0.5 ± 0 | 243 ± 12 | 01:02.5 | 2,833 ± 252 | |
| MANR-F | June | 76 ± 13 | 60 ± 5. | 420 ± 60 | 18:02.5 | 6,500 ± 173 | 77 ± 3[a] |
| | Nov. | 10 ± 00 | 1.3 ± 0.8 | 363 ± 12 | 45:02.5 | 1,500 ± 200 | |
| RCW-F | June | 66 ± 16 | 68 ± 13 | 330 ± 12 | 10:02.5 | 6,767 ± 58 | 60 ± 6[b] |
| | Nov. | 10 ± 00 | 0.5 ± 0 | 283 ± 22 | 00:02.5 | 2,700 ± 400 | |
| BSG-F | June | 10 ± 00 | .40 ± 16 | 990 ± 87 | 52:02.5 | 7,133 ± 58 | 38 ± 11[c] |
| | Nov. | 80 ± 35 | 4 ± 3.5 | 733 ± 55 | 33:02.5 | 2,700 ± 400 | |

**Note:**
TOC stands for total organic carbon (values ± SD). There is low variability within the treatments' initial PHC concentrations and they are statistically different from each other. Reduction presents the percent PHC removed and the lowercase letters denote statistical rank (mixed model, $p < 0.05$).

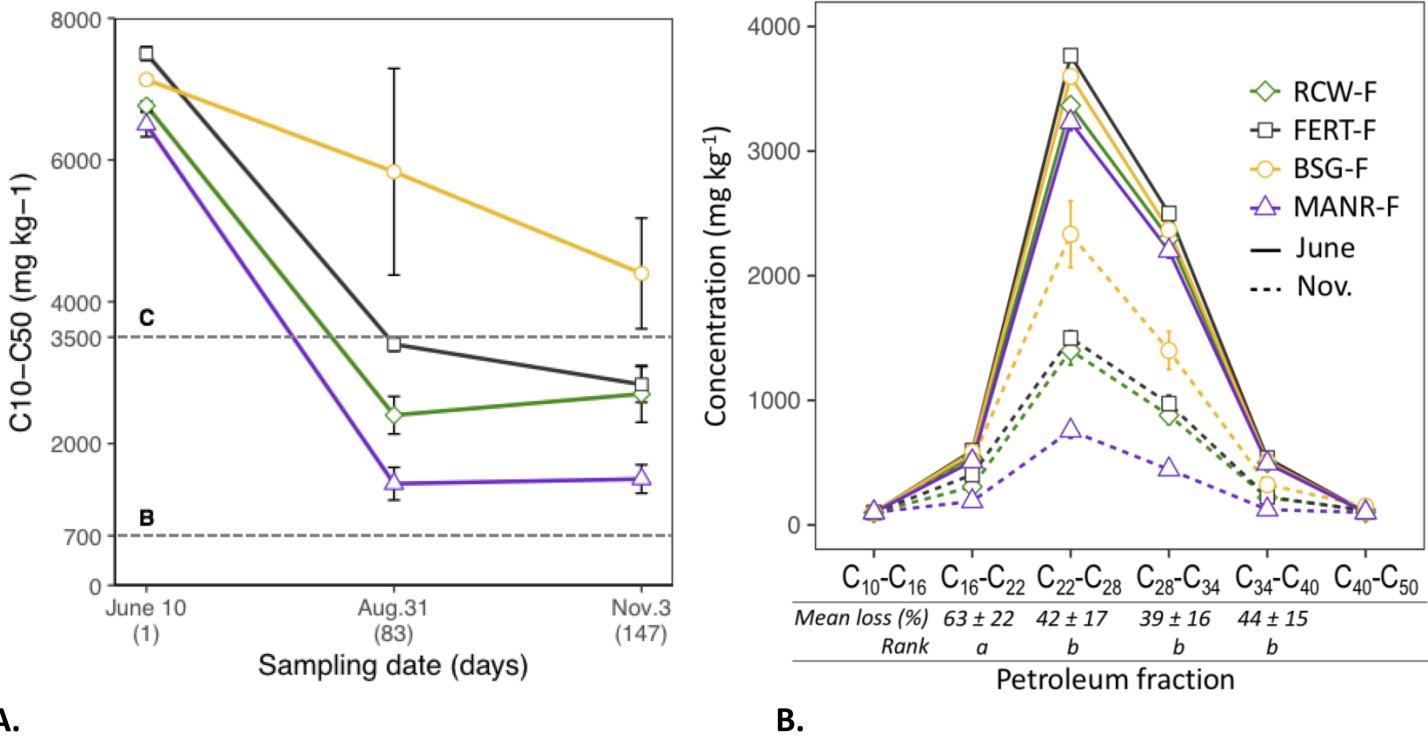

**A.**          **B.**

**Figure 2 Petroleum hydrocarbon concentrations.** (A) Concentrations of engine oil fraction $C_{10}$ to $C_{50}$ over 147 days. The lines at 3,500 and 700 mg kg$^{-1}$ indicate the Quebec environmental guidelines for Commercial & Industrial sites ("C") and Residential & Recreational sites ("B") respectively (*Beaulieu, 2016*). (B) Concentrations of the six petroleum hydrocarbon fractions monitored in the soil. The dotted lines represent the last sampling event (November). Mean loss compares the percent decreases in the different fractions. The rank is Tukey HSD (Bonferroni correction applied for multiple comparisons, after a mixed model analysis ($p < 0.05$)). For both figures, reported errors are standard deviation (SD, $n = 3$).

experiment (Fig. 2A). There is little variability within each treatment for the initial PHC data before RFMs addition ($\pm57$ to $\pm173$ mg kg$^{-1}$, which represents a variability range of 0.8–2.7%), indicating a homogeneous initial distribution of the engine oil in the soil. However, PHC concentrations were different for all treatments at the start ($p < 0.05$), which is likely due to uneven dilution from the different RFMs since they were added on a volume basis and PHC concentrations are expressed on a mass basis (Table 1). MANR-F was significantly more effective at driving a PHC reduction (77% $\pm$ 3% reduction) in the soil than all other treatments (BSG-F: 38% $\pm$ 11%, RCW-F: 60% $\pm$ 6% and FERT: 62% $\pm$ 4%) (mixed model, $p < 0.05$). On August 31, while the platform's usual FERT was fluctuating around the Commercial & Industrial sites' regulatory guideline (3,500 mg kg$^{-1}$), MANR-F was well below it and approaching the Residential & Recreational guideline (700 mg kg$^{-1}$) (Fig. 2A). The analysis of engine oil by fraction revealed the lightest ($C_{10}$–$C_{16}$) and heaviest ($C_{40}$–$C_{50}$) fractions were either close to or below the detection limit (100 mg kg$^{-1}$) and were excluded from further analysis. The next lightest fraction ($C_{16}$–$C_{22}$) decreased significantly more (63% $\pm$ 22%) than the rest of the fractions ($C_{22}$–$C_{28}$, $C_{28}$–$C_{34}$, $C_{34}$–$C_{40}$), which were removed at equivalent rates (42% $\pm$ 17%, 39% $\pm$ 16% and 44% $\pm$ 15%, respectively), despite differences in molecular size (Fig. 2B).

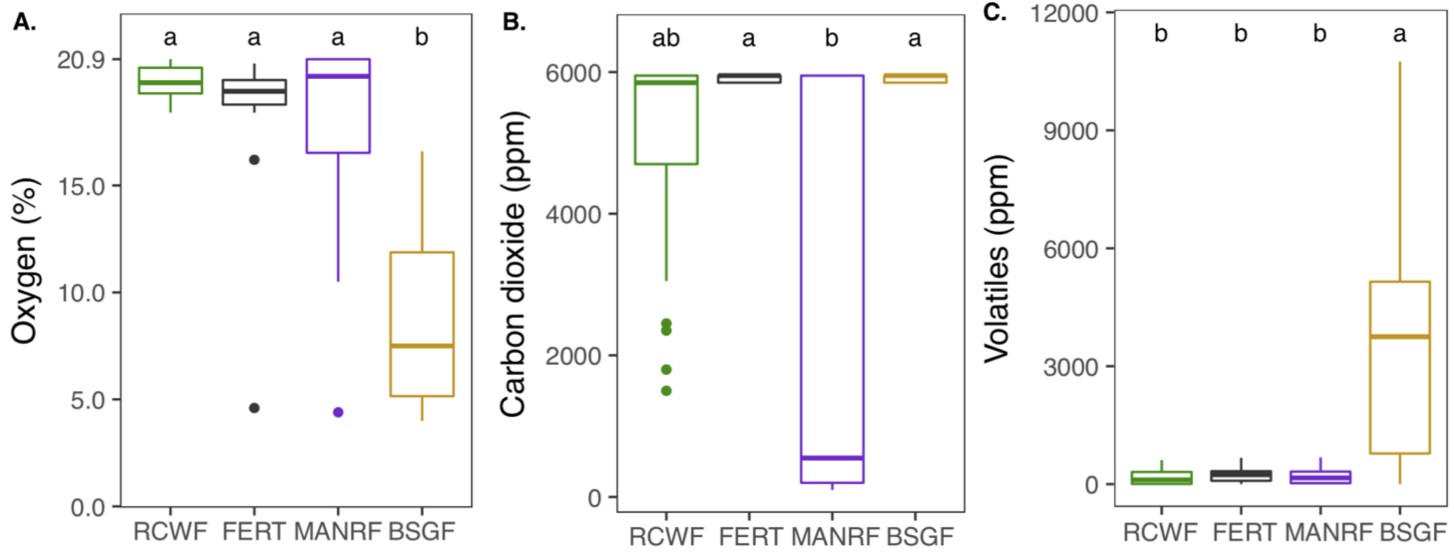

**Figure 3  Gas concentrations in the center of the treatment bins measured seven times over 145 days.** (A) Oxygen (%). (B) Carbon dioxide (ppm). (C) Volatile organic compounds (ppm). Error bars represent SD. Lower case letters denote statistical ranking (Kruskal–Wallis, $p < 0.05$, Bonferroni correction applied for multiple comparisons. $n = 21$).

## Outgassing

There were notable differences in the gas emissions monitored among the different treatments, despite high variability within the data (Fig. 3). With a mean of 7.9% ± 3.6%, BSG-F contained significantly less $O_2$ than the other treatments, which all showed averages between 18% and 19% ($p < 0.05$). $CO_2$ emissions by FERT and BSG-F were always near the upper detection limit of the equipment (6,000 ppm), while MANR-F was the treatment with the lowest mean emission (2,695 ± 2,871 ppm) ($p < 0.05$). VOCs were very low in all treatments (means below 250 ppm), except for BSG-F (3,636 ± 3,318 ppm), which released significantly more than all other treatments ($p < 0.05$). The VOCs emitted in this case could likely be methane from the breakdown of organics in waterlogged pockets of the cylinder which had become anoxic.

## Community-level physiological profiling

The global metabolic activity of the plates (AWCD) showed that the soils with amendments were more active than the FERT by the third day and by the fourth day, BSG-F had more AWCD than all others (Fig. 4A) (mixed model, $p < 0.05$). All treatments followed a sigmoidal shape, presenting a standard bacterial growth curve. The CI approach also revealed that FERT-F was less active than the soils amended with local RFM. There were differences in carbon usage between treatments. For example, BSG-F was able to use polymer carbon sources more effectively than the other treatments and FERT-F used carbohydrates less effectively than all other treatments (mixed model, $p < 0.05$). Multiple significant differences were also noted within the usage of the carbohydrates, where bacteria in RCW-F were able to actively use more C sources, more rapidly than all other treatments. There was less functional richness in FERT (240 active wells out of 288) than in MANR-F (270), RCW-F (270) and BSG-F (273). Finally, the PCA indicates a

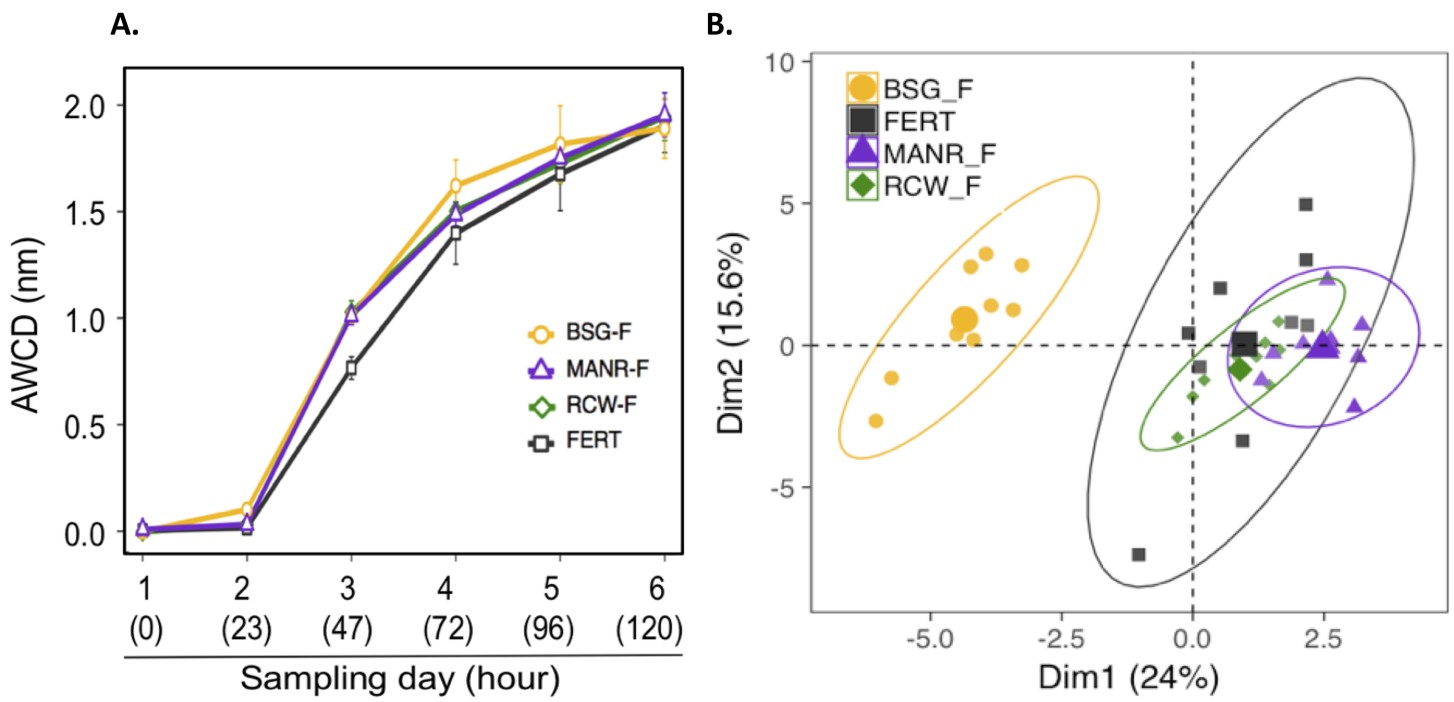

**Figure 4 Ecoplate results.** (A) Average well color development (AWCD) (nm) over 6 days for soil from the four treatments. Error bars represent standard deviation. (B) PCA: Principal component analysis of the consumption of the 31 carbon sources in relation to the four treatments.

clear separation between BSG-F and the other three treatments, which form over-lapping but clear clusters (Fig. 4B). This confirms the different soils' bacterial community's catabolic activity fingerprint.

## DISCUSSION

### CLPP and soil properties

Community-level physiological profiling conducted with the Ecoplates™ indicated that the biological amendments increased the metabolic activity and functional richness of the soils compared to the fertilizer control (FERT). No well-defined links between PHC-degradation efficiency and Ecoplates-derived metabolic activity were observed. Nonetheless, there were clear clusters in the PCA defining the catabolic capabilities of the soils based on amendment. Microbial activity is optimal between 20 and 40 °C for petroleum remediation and slows when soil temperatures drop close to 0 °C (*Battelle & NFESC, 1996*). In this project, overall higher temperatures were not linked to higher rates of contaminant removal. Other authors have found that temperature had a great influence on the reduction of petroleum concentrations (*Van Gestel et al., 2003*), or that higher temperatures could lead to increased volatilization (but unlikely with heavy PHC) (*Namkoong et al., 2002*). In our case, BSG-F exhibited the highest temperatures, but was the least effective at PHC-removal. This was most likely due to the heat generated during fermentation (anaerobic breakdown) of the BSG. Soil pH is influenced by microbes, and can also be influenced by their activity (*Nwankwegu & Onwosi, 2017*). Over the course

of this experiment, pH remained stable in all treatments, except in the BSG-F. Other studies have reported pH variations in most treatments; this difference in our work could be linked to the soil's pH buffering capacity or the large-scale set-up, which has different dynamics than microcosms (*Nwankwegu & Onwosi, 2017*; *Prince, 2015*). Moisture is an important factor for bioremediation: concentrations too low inhibit microbial activity, and excessive moisture levels can promote hypoxic conditions that are less conducive to rapid biodegradation of PHC (*Khan, Husain & Hejazi, 2004*). A moisture content of 50–70% of field capacity is often cited as optimal (*Agency for Toxic Substances and Disease Registry (ATSDR), 1999*; *Khan, Zytner & Feng, 2015*). In the sandy soil used in the trials, 70% of the field capacity corresponds to about 15% moisture on a mass basis (g g$^{-1}$). Field capacity varies with soil type and organic matter in soil can improve water retention (*Brady & Weil, 1996a*). The various local organic amendments used in this project clearly demonstrated this point, with the strong correlation between non-PHC carbon in the soil and moisture levels, and with the different watering requirements of the treatments. FERT had virtually no organic matter and required moisture addition throughout the project. RCW-F required watering the first two events, but not the third, indicating that the wood had started to retain water in the soil. BSG-F retained moisture slightly above levels for optimal degradation (20% ± 1.7% (g g$^{-1}$) at the start), but the aspect of most concern was that this high moisture was coupled with a fine particle and compact amendment structure which limited air circulation in the bins, fostering hypoxic conditions which are not ideal for rapid PHC degradation. Further, the BSG supplied soil microbes with an abundance of easily accessible food which may have been easier to target than the engine oil. Finally, MANR was the optimal RFM input for moisture since MANR-F did not require any watering throughout the project, while maintaining values close to 70% field capacity. Reduction in soil management operations can be advantageous in large-scale settings.

## Nitrogen balance and engine oil removal

In this study, nitrogen came from two main sources: an inorganic fertilizer and the three RFMs (soil nitrogen content was negligible). No nitrogen was added during the course of the study. We had aimed to keep the soils under aerobic conditions where volatile forms of N are not normally emitted, hence total nitrogen should have remained stable throughout the monitoring. It did, with the exception of BSG-F where a nearly 25% drop in nitrogen occurred. Other than heterogeneous distribution in soil which can skew the results, total nitrogen loss in soils may be related to leaching of soluble forms (although bins were largely protected from rainfall) or a denitrification process resulting in the formation of nitrogen gas and its loss to the atmosphere. This last process may indicate conditions in soils that are too low in oxygen. The reduced forms of nitrogen ($NH_4^+$ and $NH_3$) present in the BSG-F soil in November support the presence of sub-optimal oxygen levels since these reduced forms of nitrogen are usually observed when the soil is deficient in oxygen. Whereas, when the soil is well aerated, nitrogen is found in oxidized forms such as nitrate and nitrite (*Brady & Weil, 1996b*). There was an initial input of ammonium as diammonium phosphate in the inorganic fertilizer, but this was not

detected in the BSG bins at the beginning of the experiment. The concentrations detected in November were likely mineralized from the organic matter in the amendment and remained in this form due to lack of oxygen for conversion to oxidized forms ($NO_3^-$ and $NO_2^-$) by the nitrification process. Reduced forms of nitrogen are most prevalent under hypoxic soil conditions (*Britto & Kronzucker, 2002*) and effective aliphatic PHC-degradation usually occurs under aerobic conditions (*Khan, Husain & Hejazi, 2004*).

Larger PHC molecules were expected to be biodegraded at slower rates than the lighter ones, but this was not generally the case in our study (*Battelle & NFESC, 1996*; *Khan, Husain & Hejazi, 2004*). The $C_{16}$–$C_{22}$ fraction was indeed removed faster than the other fractions monitored, but the decrease of the other three fractions (between $C_{22}$ and $C_{40}$) was statistically equivalent ($p > 0.05$). PHC can move out of soil through volatilization, but other authors found mainly fractions below 12–16 carbons to be volatile while heavier compounds were mainly biodegraded (*Namkoong et al., 2002*; *Gallego et al., 2010*). Bacteria usually reduce PHC molecules down to $CO_2$ (*Brady & Weil, 1996c*). After 82 days of the different treatments, MANR-R had significantly decreased PHC concentrations and was approaching the Residential and Recreational guideline (700 mg kg$^{-1}$), while the LTU's usual FERT treatment was fluctuating around the Commercial and Industrial guideline (3,500 mg kg$^{-1}$) of the Quebec government regulations. From this point on to the end of the experiments all treatments plateaued, and little degradation took place in the last 65 days. Since varying levels of carbon from the PHC remained in the soil, it would be possible to optimize the treatments to stimulate degradation beyond the plateaus. This could be done through better control over parameters such as aeration, moisture and nutrients, and using known correlations for estimating petroleum biodegradation rates in soils, which are especially useful for scaling-up experiments (*Khan, Zytner & Feng, 2015*).

BSG-F did not exhibit the same plateau but was far less effective than the other treatments for engine oil removal (38% ± 11%). Other authors have found BSG to be an effective soil amendment. In a phytoremediation experiment using the plant *Jatropha curcas*, *Agamuthu, Abioye & Aziz (2010)* found that when BSG was added to a soil contaminated with 2.5% and 1% of lubricating oil, the degradation rates of the PHCs were 89.6–96.6% respectively. *Oruru (2014)* demonstrated not only the efficiency but also the sustainability of BSG as an amendment to treat diesel-contaminated soil. The reduced efficiency of the BSG-F in this study appears to be linked to the soil structure and excess moisture which restricted aeration. Despite its lower PHC-removal performance in this project, the application of BSG could be further considered for the remediation of soils with low nutrients and low water retention capacities, as long as aerobic conditions are maintained to minimize nitrogen loss and VOC output. Currently, farmers supplement their animals' regular feed with BSG, but rapid disposable is primordial for breweries because it ferments rapidly causing serious odor problems. This leads many city-based breweries to send their BSG to the landfill (*Santos et al., 2003*).

RCW-F was equivalent to FERT for PHC-removal and its measured contributions within this project are limited to improvements of soil parameters (moisture and microbial activity). MANR-F was unequivocally the most successful soil amendment in this project. It led to the most PHC-reduction in the shortest time period. From a soil clean-up

facility's perspective, this represents a faster turnaround of the soil and potential increases in profitability. MANR-F maintained adequate moisture and oxygen levels, while releasing the least amount of $CO_2$, which is interesting in a global climate change context. The bacterial degradation of PHC emits $CO_2$ and since the most PHC-removal took place in this treatment, it should have released more than the rest. Since it was not the case, we hypothesized a form of carbon sequestration in the microbial biomass, but this was not tested further.

## Cost of RFM and opportunities for circular economy?

Within the scope of this project, MANR was available nearby and free of cost, but in other regions such manure may be sold at a higher price or used directly on adjacent farmlands, potentially reducing the application of our results to some regions. RCW did not significantly increase remediation speed compared to fertilizer alone, but it promoted long term moisture retention, which can alleviate the management of biopiles through reduced watering needs. BSG led to less PHC removal than the LTU's usual fertilizer treatment by limiting aeration due to excess moisture. In retrospect, the addition of 30% BSG for this soil was excessive. We hypothesize that a smaller proportion of the dense and nutritious BSG, coupled with the addition of a bulking agent such as RCW could create favorable PHC remediation conditions at an advantageous cost. We conclude that there is no one-size-fits-all approach for the use of RFMs, and each remediation site will need to evaluate the RFMs available in their respective region.

In their Vision 2050 project, the World Business Council for Sustainable Development envisions making the concept of waste obsolete as a normal business practice (*World Business Council for Sustainable Development (WBCSD), 2010*). This research highlights a particularly interesting opportunity for soil treatment industries to actively participate in this vision, by up-cycling degraded "waste soil" through remediation and introducing it into a circular economy loop. Further, RFM are present across the globe and can offer low-cost amendments to boost remediation efficiency while reducing treatment time.

## CONCLUSION

Soil remediation industries can benefit from the addition of RFM in their operations, while diverting RFMs from landfill, thereby contributing to waste diversion objectives set by European and North American governing agencies. Each locality has regulations to limit the quantity of material that can be incorporated into contaminated soil, to avoid excessive dilution. Overall, RFM's main functions are improving moisture retention and soil structure, contributing nutrients and a greater diversity of microorganisms, and to enhance the growth matrix for microbial life in the soil, which can, in some cases, enhance PHC remediation as compared to fertilizers alone. These conclusions are unsurprising in the sense that adding organic matter to contaminated soil has been shown to foster bioremediation in the past, but we view the process undertaken in this project as another step toward more partnerships between academia and industry, in a manner that incorporates considerations of the broader environmental impacts of bioremediation work. The circular economy objective of reducing pollution and waste as much as possible served as a guiding principle to sourcing the three RFM amendments within 100 km of the

study site. Due to performance pressures, inorganic fertilizer use was maintained in the scope of this project, but future work should include the exploration of RFMs and processes which may allow for a move away from this traditional approach, which requires resources that are synthesized by high-energy processes and sourced from non-renewable resources (such as P). Overall, choice of the RFM treatment should be made considering multiple factors such as treatment effects, cost, time, accessibility by users, and sustainable supplying, to fall into the concept of a circular economy.

## ACKNOWLEDGEMENTS

Thank you Pierre Yves Cardon for your invaluable help with the Ecoplates work. Thank you to À la Fût microbrewery for providing the BSG and helping with the logistics around its transport. Thank you to Stéphane Lambert and Martine Sanchez for welcoming this project on your platform. And last but not least, thank you to Yves Tourangeau for your enthusiastic dedication to the project and strong field work!

### Funding

The research was funded by the Canada Research Chair in Global Change Ecotoxicology to Marc Amyot and NSERC CREATE Mine of Knowledge program to Kawina Robichaud. Akifer and SolNeuf provided land space, PHC analysis expenses, and in-kind: specialists, machinery, fertilizers and logistical support. The funders had no role in study design, data collection and analysis, decision to publish, or preparation of the manuscript.

### Grant Disclosures

The following grant information was disclosed by the authors:
Canada Research Chair in Global Change Ecotoxicology.
NSERC CREATE Mine of Knowledge program.
Akifer and SolNeuf provided land space, PHC analysis expenses, and in-kind: specialists, machinery, fertilizers and logistical support.

### Competing Interests

Marc Amyot is an Academic Editor for PeerJ. This work was conducted by a Ph.D. student, Kawina Robichaud, as part of a governmentally-funded internship program in a private company (Akifer), aimed at better training graduate students. Akifer provided the experimental units, funded some of the analyses and provided expert advice. Akifer do not benefit in any way from this publication and they simply agreed to it for the benefit of the graduate student who will use this paper as part of her thesis. Miriam Lebeau and Sylvain Martineau are employed by Akifer Inc.

### Author Contributions

- Kawina Robichaud conceived and designed the experiments, performed the experiments, analyzed the data, prepared figures and/or tables, authored or reviewed drafts of the paper, approved the final draft.

- Miriam Lebeau conceived and designed the experiments, performed the experiments, analyzed the data, contributed reagents/materials/analysis tools, authored or reviewed drafts of the paper, approved the final draft.
- Sylvain Martineau conceived and designed the experiments, analyzed the data, contributed reagents/materials/analysis tools, authored or reviewed drafts of the paper, approved the final draft.
- Marc Amyot conceived and designed the experiments, analyzed the data, contributed reagents/materials/analysis tools, authored or reviewed drafts of the paper, approved the final draft.

## Data Availability

The raw data are provided in the Supplementary Files. The raw data include gas levels, ecoplate results and concentrations of PHC used to prepare the different figures and tables.

## Supplemental Information

Supplemental information for this article can be found online at http://dx.doi.org/10.7717/peerj.7389#supplemental-information.

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
