# Peer review of "Bioremediation of engine-oil contaminated soil using local residual organic matter"

_PeerJ, doi:10.7717/peerj.7389_

## Round 0.1 · original submission · Minor Revisions

Thank you for submitting your work in our journal and very sorry for the delay. I think your manuscript sounds very solid and should benefit from the comments from the reviewers, in particular the comments on the result validity by reviewer 2. Regarding the literature proposed by reviewer 1, I suggest you evaluate if it inclusion really makes sense. From my side I am not entirely sure the references proposed here fit.

·

Basic reporting

In this work, authors have carried out a study on bioremediation of engine-oil contaminated soil using local residual organic matter. The study design considered four treatments notably; ramial chipped wood (RCW), horse manure (MANR), brewers ‘spent grain (BSG), and inorganic fertilizer (FERT). Authors sufficiently used many analytical parameters to support the study. This manuscript would be recommended for publication, however, I would recommend some minor corrections prior to acceptance.
Comments
Abstract
(1) L21 – 22, authors should consider using either “framework of circular economy” or “circular economy’s framework”

Introduction
(2) In L57 – 58, some of the citations used by authors are way too old and not recent enough to warrant a novel study.
(3) Authors are invited to consider the following recommendations wherein strength of locally generated fertilizing materials has been recently demonstrated.
Onwosi et al., 2018
Onwosi, C.O., Nwankwegu, A.S., Enebechi, C.K., Odimba, J.N., Nwuche, C.O., Igbokwe, V.C., 2018. Bioremediation of Soil Contaminated with Diesel Using Inorganic Nitrogen Sources: Incorporating nth- Order Algorithm in the Evaluation of Process Kinetics. Soil and sediment contamination. 1 – 9. https://doi.org/10.1080/15320383.2018.1423023.
Onwosi et al., 2017
Onwosi, C.O., Odibo, F.J.C., Enebechi, C.K., Nwankwegu, A.S., Ikele, A.I., Okeh, O.C. 2017. Bioremediation of diesel-contaminated soil by composting with locally generated bulking agents. Soil and Sediment Contamination. 26(4), 438–456.

Nwankwegu et al., 2016a

Nwankwegu, A.S., Onwosi, C.O., Orji, M.U., Anaukwu, C.G., Okafor, U.C., Azi, F., Martins, P.E., 2016a Reclamation of DPK hydrocarbon polluted agricultural soil using a selected bulking agent. J Environ Manag. 172, 136 – 142.
(4) L62. Circular economy’s framework. Revise.
(5) L82 – 84 essentially needs citation (s). The following or another would be appropriate;
Nwankwegu et al., 2016b
Nwankwegu, A.S., Orji, M.U., Onwosi, C.O., 2016b. Studies on organic and in-organic biostimulants in bioremediation of diesel-contaminated arable soil. Chemosphere.162, 148 – 156.

Nwankwegu et al., 2017

Nwankwegu, A.S., Onwosi, C.O., Azi, F., Azumini, P., Anaukwu, C.G., 2017. Use of rice husk as bulking agent in bioremediation of automobile gas oil impinged agricultural soil. Soil Sediment Contam. 26, 96 – 114.
(6) Consider removing Hattab et al., 2015 in L108 its appearance in L107 is already an ideal. This should be effected where applicable in the entire manuscript.
(7) In L116, authors should replace the verb “has” with have.
(8) The citation “Abioye (2011) as it appeared in L119 should be revised.
(9) L127 – 128, what are those legal regulations? A mention of two or more would sufficiently increase the strength of assertion.
(10) Authors considered more than one treatments in the study so the word “between” as used in L138 should be replaced with “among”

Experimental design

Materials and methods
(11) In L156 according to authors ‘explanation, are they part of the RFM not locally sourced?
(12) In L236, it is not clear if the authors stated how the protocol is done rather than what was done in the present study. Saying “extraction is done” is like stating a general procedure instead of what was done.
(13) L249 – 250, it is noteworthy for attention why the authors chose the use of old citations, whereas more recent studies which discussed the similar approach abound. For example;
Nwankwegu and Onwosi, 2017
Nwankwegu, A.S., Onwosi, C.O., 2017. Bioremediation of gasoline contaminated agricultural soil by bioaugmentation. Environ. Technol. Innovat. 7, 1 – 11.
(14) In the whole methodology, it is clear that authors did not monitor the pH fluctuations in the different amendments. Studies have shown that no bioremediation approach is 100% efficient because of different metabolite that maybe produced as bioremediation event lasts. It is also demonstrated that pH in different microcosms under this condition is usually the function of dominant metabolite arising from the amendments. It is actually surprising that the pH of the inorganic fertilizer amended options did not significantly decrease relative to the organic material amended systems but remained stable throughout the incubation. Technically, one may wish to know why? For examples; Orji et al., 2012 (Malaysian Journal of Microbiology), Nwnakwegu et al., 2018 (Environmental Technology), Nwankwegu et al., 2016 (Chemosphere), etc. have attributed substantial contribution to pH dynamics. The pH is therefore as important as temperature and other physicochemical parameters. This could be the first report of pH stability under different treatments.
(16) What do the authors mean by the acronym PCN in L299? Is it similar to PHC as previously used in the previous sections? If yes then it is crucial to maintain consistency in the acronyms if no then initial definition would be more compelling.
(17) In L300, authors may not necessary introduce citations in the result section since it is separated from the discussion and not results and discussion. If this is a statement that needs to be backed by citations then I suggest you lift them to the discussion section.
(18) The word “hovering” in L319 needs to be revised. Consider “was within the commercial & Industrial sites’ regulatory guideline (3500 mg kg-1) instead of hovering. The personification is way too animate.
(19) In L328, replace “between” with “among”
(20) In L338, initial definition of AWCD is important prior to subsequent acronym. AWCD could mean Average Well Color Development.
(21) It is further more surprising that considering what happened from L338 – L351, pH stability was reported. I would therefore challenge the authors to show previous reports of pH stability under similar scenarios. If not I kindly request authors to revise and include pH dynamics as it certainly would affect microbial amensalism and bioremediation efficiency.
Discussion
(22) The study did not consider metabolomics evaluation of PHC degradation hence the statement in L356 –L357 lacks authority
(23) Consider revising this word “hovering” in L415
(24) L422 –L427, what kind of citation is this e.g. Agamuthu et al. (2010)et al.
423 (2010)?
(25) L432 – L433, statement not clear. Revise
(26) Consider changing least amount of time to shortest time in L438.
(27) What do the authors actually mean by the use of the acronym PCH in L441 without initial definition?
(28) L450 – L451, what about cost?
Conclusion
(29) L465, consider changing the infinitive” to source” to “to sourcing”

Validity of the findings

I strongly advocate that the study demonstrated significant validity of finding while I want the authors to further address the effect of pH on PHC removal efficiency

Additional comments

Generally, the whole work especially the conclusion showed great strength but I sincerely recommend more recent citations. This would do pretty much work on the overall strength and novelty.

Reviewer 2 ·

Basic reporting

Line 55-57, should these two sentences being switched? "Several studies demonstrate the potential of RFM as bioremediation agents..However, the MELCC (2014) reports that only 8% of salvaged RFM were used on degraded sites and usage on PHC-contaminated soils was not mentioned. ". So it helps emphasize the importance of this work. Also, authors should add more details of how those studies demonstrate the potential of RFM as bioremediation agents by not actually using RFM on PHC-contaminated soils?

Line 67, what about US? and Canada? I think the whole world is working on circular economy now.

Line 78, typo, "to be covered..."

Line 82, the first sentence can be more specific. appropriate nutrient input in terms of amount and time, right?

Line 86, what is "this parameter"? total N, or different forms of N (inorganic, organic)?

Line 138, "among treatments".

Experimental design

Line 143, "and operated by.." no need using "but"

Line 146, why mentioning "Fe and P" but not other elements (e.g. N, P)? I don't expect to see those because the context of "importance of Fe and P" was not introduced in the Introduction.

Line 177, omit "in"

Line 170-187, if authors could create and add a diagram showing the "flow" of this method, it would help improve readers to understand those steps.

Line 193-194, change to "Samples were taken from 30 cm and 60 cm depths in the first hole, and from 60 cm and 90 cm depths in the second hole"

Line 208-245, this section is a little long, probably have several paragraphs instead of just one. Also, group sentences about ammonia, nitrate and nitrite ions together because they are all about nitrogen.

Line 266, "among treatments".

I appreciate those clear sentences with enough details in the statistical section from the authors!

Validity of the findings

Line 300, I did not expect to see references in the Result section, if there is, it is more like a discussion point.

Line 301-302, "The horse manure (MANR) contained the most potassium, and ramial chipped wood (RCW) contained the least nutrients overall" this sentence is inappropriate, authors comparing "MANR's potassium" vs. "RCW's all nutrients"?

Line 302, "available nitrogen" is nitrogen in forms that ready to be uptake by plants. I assume NH3 is not one of them?

Line 305, how did authors define the nitrogen loss is "low"? within 10%? or this is comparing to the loss of nitrogen for BSG-F?

Line 342, "some differences" this wording is not professional, edit it

Line 360, could be 0 instead of zero

Line 361, omit "seemed to", if that's their results not from their discussion (assumption)

Line 366, it might be interesting to develop a parameter "temperature/moisture" to see the combine effects of temperature and moisture to the treatment since authors have examined their impacts individually.

Line 407, the choice of degrading which PHC molecules by bacteria might depend on the total energy source they need. I guess the assumption of "larger PHC molecules degraded at slower rates" is assuming the energy source (or C:N balance?) is not enough?

Line 418, what kind of "optimize the treatments"? adjust the temperature or moisture? or changing to a more efficient treatment?

Line 470-481, this paragraph of discussing cost and accessibility of some treatments is very valuable, but might be better to move up to section of line 445? In the Conclusion section, the statement should be more broad saying "Choice of the treatment should be considering multiple factors treatment effects, cost, time, accessibility by users, sustainable supplying, etc, to fall into the concept of circular economy"?

Additional comments

This paper examined the bioremediation of engine-oil contaminated soils using residual fertilizing matter additions, and compared three types of matters (ramial chipped wood, horse manure and brewer spent grain) with a baseline fertilizer treatment. It measured multiple variables including VOCs, O2, CO2, PHC fractions, Microbial community level physiological profiling, etc. It reported that addition of horse manure had the best performance, but other types of residual fertilizing matters might also have the potential (e.g. smaller proportion of the dense and nutritious BSG with ramial chipped wood). I found this paper is very valuable to the society.

---

## Round 0.2 · accepted · Accept

Thanks for taking into account the comments from both reviewers which address the issues. Congratulations!